# Generating dermatopathology reports from gigapixel whole slide images with HistoGPT

Manuel Tran [1,2,18], Paul Schmidle[3,18], Ruifeng Ray Guo [4], Sophia J. Wagner [1,2], Valentin Koch [2,5], Valerio Lupperger [6], Brenna Novotny [7], Dennis H. Murphree[8], Heather D. Hardway[8], Marina D'Amato [9], Judith Lefkes [9,10], Daan J. Geijs[9,10], Annette Feuchtinger[11], Alexander Böhner[12], Robert Kaczmarczyk [12], Tilo Biedermann [12], Avital L. Amir[13], Antien L. Mooyaart[14], Francesco Ciompi [9], Geert Litjens [9,10], Chen Wang [7], Nneka I. Comfere[8,15], Kilian Eyerich [3,19] ✉, Stephan A. Braun [16,17,19] ✉, Carsten Marr [1,5,19] ✉ & Tingying Peng[1,2,19] ✉

Histopathology is the reference standard for diagnosing the presence and nature of many diseases, including cancer. However, analyzing tissue samples under a microscope and summarizing the findings in a comprehensive pathology report is time-consuming, labor-intensive, and non-standardized. To address this problem, we present HistoGPT, a vision language model that generates pathology reports from a patient's multiple full-resolution histology images. It is trained on 15,129 whole slide images from 6705 dermatology patients with corresponding pathology reports. The generated reports match the quality of human-written reports for common and homogeneous malignancies, as confirmed by natural language processing metrics and domain expert analysis. We evaluate HistoGPT in an international, multi-center clinical study and show that it can accurately predict tumor subtypes, tumor thickness, and tumor margins in a zero-shot fashion. Our model demonstrates the potential of artificial intelligence to assist pathologists in evaluating, reporting, and understanding routine dermatopathology cases.

Histopathology stands as the clinical gold standard for diagnosing a wide range of conditions, including malignant cancers and inflammatory diseases[1]. It involves the examination of tissue samples under a microscope by pathologists who follow strict guidelines to ensure accurate and consistent results[2]. Their observations are summarized in pathology reports, which are essential for treatment decisions and communication among clinicians. However, generating these reports is time-consuming, labor-intensive, and non-standardized[3]–resulting in delays and inefficiencies[4]. In some cases, such as basal cell carcinoma, an experienced pathologist can make a diagnosis in seconds, but it takes longer to dictate or type the findings (Fig. 1a). Automating report writing with artificial intelligence (AI) can improve efficiency, reduce errors, and help meet the growing demand for diagnostic support, allowing pathologists to focus on more complex cases.

Advanced machine learning algorithms like deep neural networks[5] are typically applied to digitized microscope slides, also known as whole slide images (WSIs). They excel at image processing tasks such as cancer classification[6], tissue segmentation[7], survival prediction[8], and biomarker detection[9]. In this context, AI is used as a tool and complement to other medical tests, rather than as a replacement for pathologists[10]. There are currently two main approaches to computational pathology. Patch-level approaches use a small portion of a WSI, ranging from 224 × 224 pixels to 1024 × 1024 pixels (called an image

---

A full list of affiliations appears at the end of the paper. ✉e-mail: kilian.eyerich@uniklinik-freiburg.de; stephanalexander.braun@ukmuenster.de; carsten.marr@helmholtz-munich.de; tingying.peng@helmholtz-munich.de

patch), to generate an output[11]. By design, these patch-level approaches ignore up to 99% of the total tissue, miss potentially diagnostically relevant areas, and cannot be applied to tasks that require the full context of the entire tissue sample (e.g., tumor thickness prediction). Slide-level approaches, on the other hand, aggregate information from all patches into a slide-level representation that can be used in downstream tasks, most notably biomarker prediction[9].

A recent direction of research is to extend the capabilities of such methods by incorporating medical text. Contrastive vision language models in pathology, such as PLIP[12] and CONCH[13], align text and images at the patch-level. They are zero-shot learners, i.e., they can solve downstream tasks for which they have not been trained (e.g., cancer subtyping). However, due to their limitations, they cannot generate a textual description of the input image. Generative vision language models such as Med-PaLM M[14], LLaVA-Med[15], or PathChat[16] can output text, but only at the patch level for small image regions up to 1024 × 1024 pixels. Thus, none of the existing medical foundation models can generate reports from an entire pathology image at full resolution, let alone from multiple images simultaneously, e.g., from a serial section.

To fill this gap, we present HistoGPT, a vision language model that can generate histopathology reports from multiple gigapixel-sized WSIs. Given multiple tissue sections from the same patient at up to 20× magnification, HistoGPT uses a vision foundation model to extract meaningful features from the images and combines them with a large language model (LLM) via cross-attention mechanisms to generate a pathology report. Each generated report describes the tissue composition, cellular subtypes, and potential diagnosis. In addition, users can interact with the model through various prompts to extract additional information such as tumor subtypes, tumor thickness, and tumor margins (Fig. 1b). The output text (Fig. 1c) is fully interpretable with saliency maps that highlight the corresponding image regions for each word or phrase in the generated text. This is achieved by training HistoGPT on a large skin histology dataset from the Department of Dermatology at the Technical University of Munich, which includes 15,129 paired WSIs and pathology reports from 6705 patients written

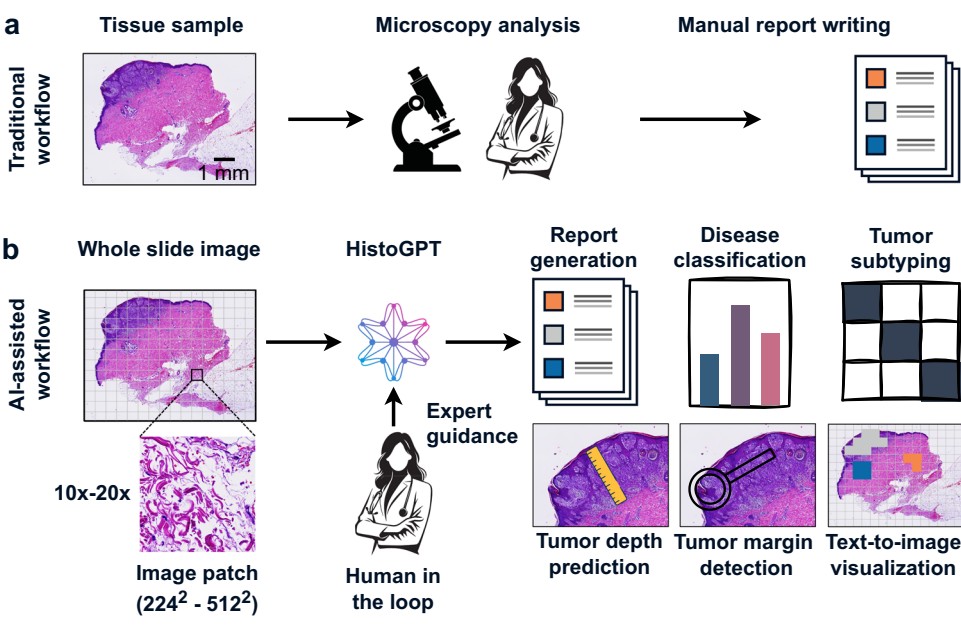

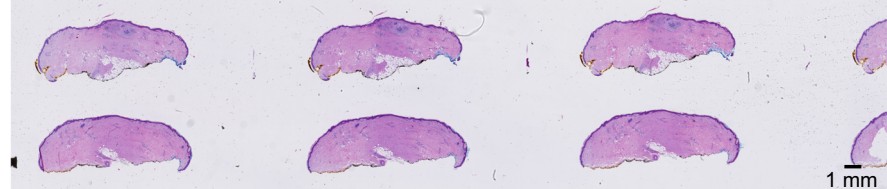

*Final diagnosis: Basal cell carcinoma. Critical findings: A superficial basal cell carcinoma is present, with a tumor thickness of 0.7 mm, in association with a cell-rich scar. The cut edges are clear. Microscopic findings: A wide punch biopsy specimen is provided. The epidermis is atrophically flattened, with a predominantly basket-weave stratum corneum. From the epidermis, there is a proliferation of basaloid tumor cell clusters into the upper dermis. Characteristic palisades are positioned in the peripheral area, with contraction artifacts and peritumoral stroma induction. A dense, plasma-rich lymphocytic inflammatory infiltrate is observed peritumorally.*

**Fig. 1 | HistoGPT, a foundation vision language model for dermatopathology.** **a** Traditionally, pathologists analyze tissue samples from patients under a microscope and summarize their findings in a comprehensive pathology report. This manual process is time-consuming, labor-intensive, and non-standardized. **b** HistoGPT generates human-level written reports, provides disease classification, discriminates between tumor subtypes, predicts tumor depth, detects tumors at surgical margins, and returns text-to-image gradient-attention maps that provide model explainability. All of this serves as a second opinion for the pathologist, who can use the output of HistoGPT as a general overview and first draft for the final report. The generated reports can also be used to fill in standardized templates, as used by some institutions, by extracting the relevant keywords. **c** An example output for a basal cell carcinoma case from our external Münster cohort. More examples can be viewed interactively at this hyperlink. Source data are provided as a Source Data file.

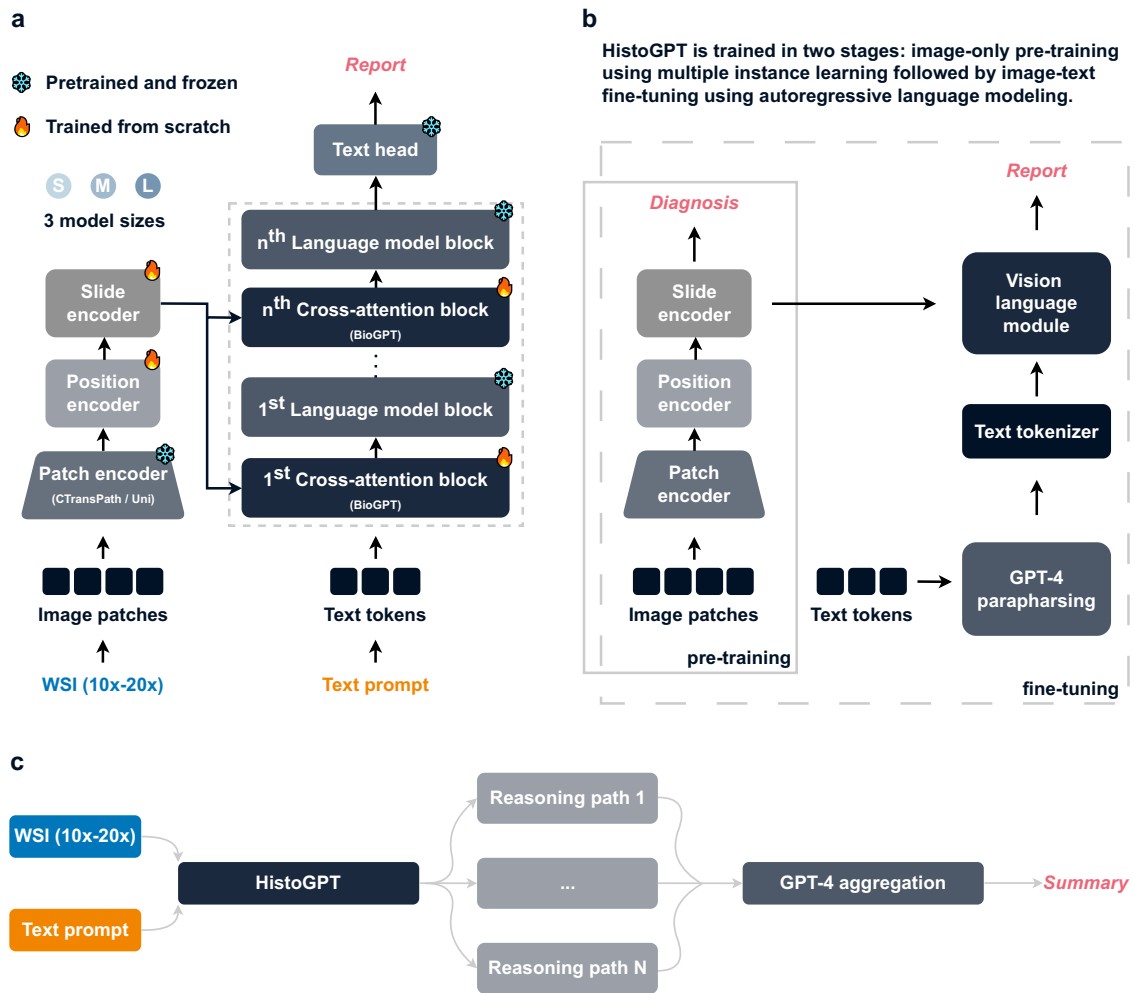

**Fig. 2 | HistoGPT simultaneously learns from vision and language to generate histology reports from whole slide images. a** HistoGPT is available in three sizes (*S*mall, *M*edium, and *L*arge). It consists of a patch encoder (CTransPath for HistoGPT-S/HistoGPT-M and UNI for HistoGPT-L), a position encoder (used only in HistoGPT-L), a slide encoder (the Perceiver Resampler), a language model (BioGPT base for HistoGPT-S, BioGPT large for HistoGPT-M/HistoGPT-L), and tanh-gated cross-attention blocks (XATTN). Specifically, HistoGPT takes a series of whole slide images (WSIs) at 10×–20× as input and outputs a written report. Optionally, users can query the model for additional details using prompts such as "The tumor thickness is", and the model will complete the sentence, e.g., "The tumor thickness is 1.2 mm". **b** We train HistoGPT in two phases. In the first phase, the vision module

of HistoGPT is pre-trained using multiple instance learning (MIL). In the second phase, we freeze the pre-trained layers and fine-tune the language module on the image-text pairs. To prevent the model from overfitting on the same sentences, we apply text augmentation with GPT-4 to paraphrase the original reports. **c** During deployment, we use an inference method called Ensemble Refinement (ER). Here, the model stochastically generates multiple possible reports using a combination of temperature, top-p, and top-k sampling to capture different aspects of the input image. An aggregation module (GPT-4) then combines the results to provide a more complete description of the underlying case. Source data are provided as a Source Data file.

by board-certified dermatopathologists. Dermatopathology covers a wide range of diseases, making it ideal for proof-of-concept studies. To validate HistoGPT, we measure the quality of the reports generated and the model's zero-shot performance in one internal and seven external cohorts across different scanner types, staining protocols, and medical procedures, such as shave biopsies, punch biopsies, or excisional biopsies. We also conduct a real-world, multi-center clinical evaluation involving six board-certified (dermato-)pathologists from three different countries. Overall, we show that HistoGPT produces clinically accurate pathology reports for the most common and routine cases.

## Results

### HistoGPT integrates vision and language to generate pathology reports

HistoGPT is a family of models with three configurations (small, medium, and large), each consisting of two components (Fig. 2a): a

vision module and a language module. The vision module is based on the patch encoder CTransPath[17] for the small and medium models, and UNI[18] for the large model. The former is a lightweight (30 million parameters) Swin Transformer[15] trained at a resolution of 1.0 micron per pixel (mpp) on over 32,000 WSIs from TCGA[16] and PAIP[19] using a semantically guided contrastive learning algorithm[20]. The latter is a much larger (300 million parameters) Vision Transformer[21] trained at 0.5 mpp resolution on over 100,000 WSIs from 22 major tissue types using self-distillation and masked modeling[22]. Our language module uses BioGPT[23], an autoregressive generative model based on the Transformer[24] decoder architecture of GPT-3[25], trained on 15 million biomedical articles from PubMed with a vocabulary size of 42,384.

HistoGPT samples image features (at 10× magnification for CTransPath, 20× for UNI) from the vision module using a slide-encoder based on the Perceiver Resampler[26], pre-trained with multiple instance learning (PerceiverMIL, Fig. 2b), and integrates its outputs into the LLM via interleaved tanh-gated cross-attention blocks (XATTN)[27]. Only

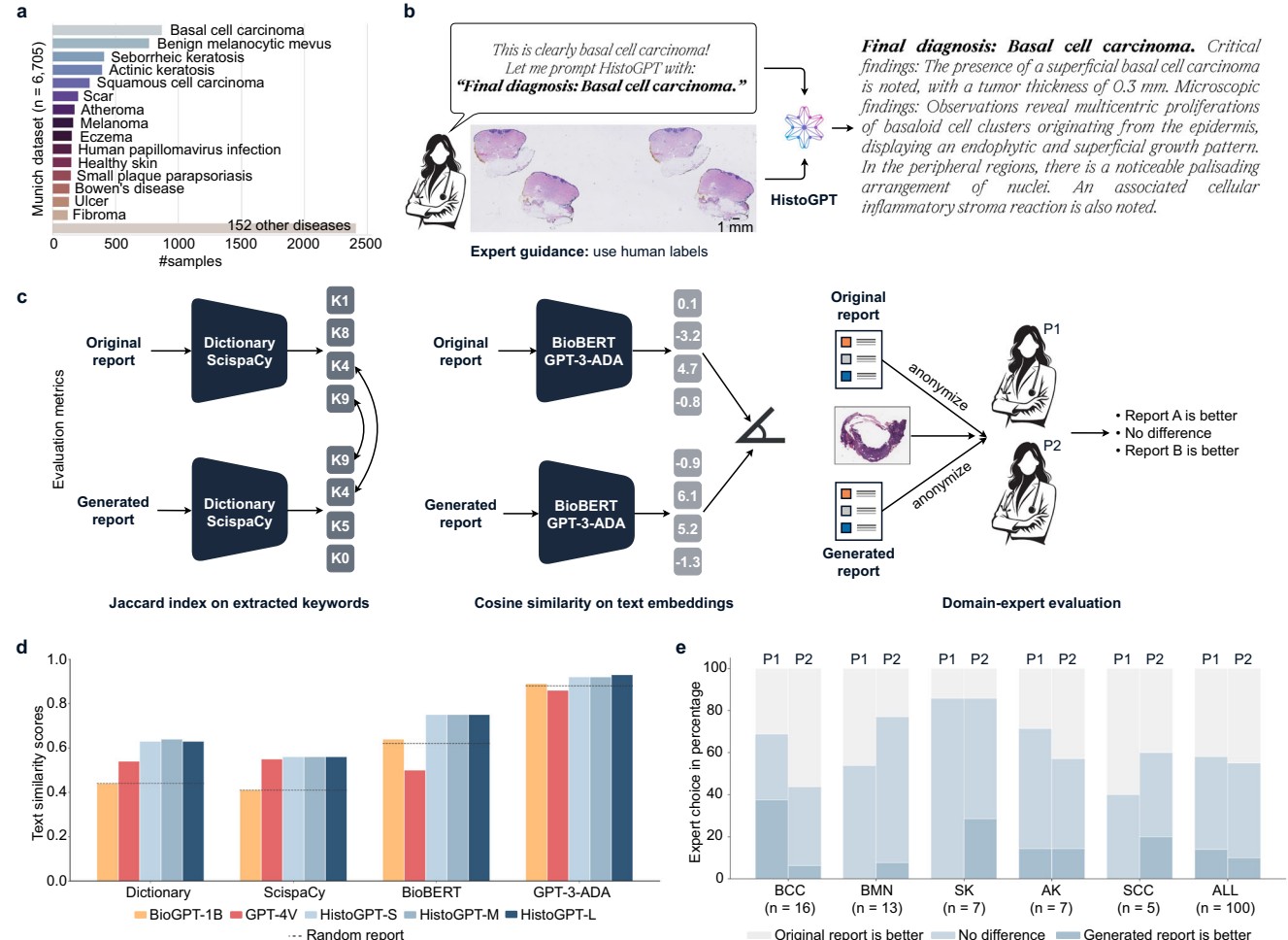

**Fig. 3 | HistoGPT generates human-level pathology reports of skin diseases.**
**a** Our internal Munich dataset is a real-world medical cohort of 15,129 whole slide images from 6705 patients with 167 skin diseases from the Department of Dermatology at the Technical University of Munich. It includes malignant cases such as basal cell carcinoma (BCC, $n = 870$) and squamous cell carcinoma (SCC, $n = 297$); precursor lesions such as actinic keratosis (AK, $n = 396$); as well as benign cases such as benign melanocytic nevus (BMN, $n = 770$) and seborrheic keratosis (SK, $n = 412$). We divided the dataset into a training set and a test set using a stratified 75/25 split at the patient level. **b** Through years of experience, pathologists are often able to make a diagnosis at first glance. Instead of writing a pathology report themselves, they can use HistoGPT in "Expert Guidance" mode by giving the model the correct diagnosis to complete the report. **c** We evaluated the performance of the model using four semantic-based machine learning metrics: (i) we matched

critical medical terms extracted from the original text with the generated text using a dermatology dictionary; (ii) we used the same technique but with ScispaCy, a scientific name entity recognition tool, as the keyword extractor; (iii) we compared the semantic meaning of the original and generated reports by measuring the cosine similarity of their text embeddings generated by the biomedical language model BioBERT; (iv) we used the same technique but with the general purpose large language model GPT-3-ADA for text embedding. **d** HistoGPT models (blue) surpassed BioGPT-1B (yellow) and GPT-4V (red) on the two text accuracy metrics, Dictionary and ScispaCy, as well as on the two text similarity metrics, BioBERT and GPT-3-ADA (see Methods for details). **e** Two independent external board-certified dermatopathologists (P1 and P2) evaluated 100 original vs. expert-guided generated reports along with the corresponding whole slide image in a randomized, blinded study. Source data are provided as a Source Data file.

these XATTN blocks are trained from scratch. In this way, we endow HistoGPT with existing visual and linguistic domain knowledge, which is critical for generating histopathology reports from entire and serial WSIs. Following Flamingo[27], we freeze the parameters of all pre-trained modules during optimization to reduce computational cost and avoid catastrophic forgetting of previously acquired knowledge (Supplementary Fig. 1). In addition, our large model uses a three-dimensional factorized position embedding function[28] to encode the x- and y-coordinates of each patch, as well as the z-coordinate indicating which slide it belongs to.

A language model predicts a probability distribution over a vocabulary: The next word in a text is selected probabilistically based on a combination of temperature, top-p, and top-k sampling. For HistoGPT, this means that once the first few words have been chosen, the outline of the report is roughly predetermined. To avoid being locked into a fixed text structure, we use an inference method called

Ensemble Refinement (ER), introduced in Med-PaLM 2[29], to randomly sample multiple reports—each focusing on slightly different aspects of the WSIs (Fig. 2c). This sampling allows us to thoroughly search the model distribution and generate a wide variety of medical reports, maximizing the likelihood of including all important observations. The general-purpose LLM GPT-4[30] is then used to aggregate all the sampled reports.

## HistoGPT generates human-level pathology reports for common diseases

Our Munich dataset is a real-world cohort consisting of 15,129 WSIs from 6705 dermatology patients with corresponding pathology reports (Fig. 3a). It contains 167 skin diseases of varying frequency and has a total size of 10 terabytes. We divided the dataset into a training set and a test set using a 75/25 split. At inference time, we prompted the model with either no diagnosis or the correct diagnosis ("Expert

Guidance"), simulating an interactive setting where a pathologist is confident in the tissue assessment but wants to leave the work of writing a draft to an AI assistant (Fig. 3b). Performance was evaluated using four semantic-based machine learning metrics and two double-blind domain expert evaluations (Fig. 3c).

HistoGPT-S, HistoGPT-M, and HistoGPT-L captured on average ~64% of all dermatopathology keywords[31] from the original pathology reports (Fig. 3d). In contrast, BioGPT-1B (a pure text model fine-tuned on our dataset using language modeling) achieved only ~44%. The state-of-the-art vision language model GPT-4V(ision)[30] improved the Jaccard index of BioGPT-1B by ~10% but still lagged behind all HistoGPT models. With Ensemble Refinement, HistoGPT-M-ER captured ~3% more terms, increasing the total coverage to ~67%. A similar trend was observed when ScispaCy[32] was used as a keyword extractor (Fig. 3d). All HistoGPT models consistently produced text with high cosine similarity to the ground truth, according to the sentence embeddings provided by BioBERT[33] and GPT-3-ADA[25] (Fig. 3d). Overall, "Expert Guidance" is the recommended modus operandi for HistoGPT, as it allows a pathologist to work interactively with the model while improving the quality of the report compared to the unguided mode (see Supplementary Table 22). We also evaluated all models using traditional syntax-based measures (BLEU-4, ROUGE-L, METEOR, and BERTscore). The relatively low syntax-based scores (see Supplementary Table 23) combined with the high semantic-based scores (Fig. 3d) support that HistoGPT does not overfit the training set by simply repeating common medical terms, like the purely text-based model BioGPT-1B, but is deeply grounded in the input image.

Text similarity analysis is only partially useful for pathology reports. Under no circumstances can it provide information as to whether the generated report is correct or not. To evaluate the generated reports from a clinical perspective, we conducted a blinded study in which we randomly selected 100 cases from our Munich test split, generated a report for each patient in "Expert Guidance" mode, and paired it with the original human-written report. The two reports were then randomly shuffled and anonymized. Two independent board-certified dermatopathologists, who were not involved in the construction or annotation of the Munich cohort, were given the original WSIs and asked to identify the report that best described each case, with the option of selecting "no difference" if both were deemed equally accurate. For the five largest diagnostic classes (basal cell carcinoma (BCC), benign melanocytic nevus (BMN), seborrheic keratosis (SK), actinic keratosis (AK), squamous cell carcinoma (SCC), see Fig. 3a), we found slight agreement between the two pathologists (Cohens' kappa = 0.09). Analyzing the results for each class separately, we found that Pathologist 1 preferred the AI-generated report or found the AI and human report similarly good in approximately 70% of the BCC cases. Pathologist 2, on the other hand, did not prefer the human report in ~80% of BMN cases. The human report for SK was not preferred by either pathologist in ~90% of cases. Across all 100 report pairs, both dermatopathologists found no difference between the AI-generated and human reports in ~45% of cases and preferred the AI-generated reports in ~15% of cases (Fig. 3d and Supplementary Fig. 2).

## HistoGPT accurately predicts diseases in geographically diverse cohorts

We extracted the predicted disease class from the generated reports to investigate whether HistoGPT predicts the diagnosis as accurately as state-of-the-art classification models based on multiple instance learning (MIL). For this purpose, we ran HistoGPT without "Expert Guidance", i.e., we simply prompted the model with the phrase "Final diagnosis:" instead of "Final diagnosis: [expert label]" and let it make a diagnostic decision on its own (Fig. 4a). Because the training set is highly unbalanced, ranging from a handful of samples in the minority classes to several hundred samples in the majority classes (Fig. 3a), established MIL methods such as AttentionMIL[34], TransMIL[35], and

TransfomerMIL[9] achieved relatively low weighted F1 scores between 0.34 and 0.48 on the Munich test set (Fig. 4b). In contrast to these MIL approaches, HistoGPT is not specialized for diagnostic prediction. Nevertheless, HistoGPT-S and HistoGPT-M attained weighted F1 scores of 0.44 and 0.45, while HistoGPT-L reached 0.48.

A challenging clinical question with a high therapeutic impact in dermatopathology is the differentiation of cancer from non-cancer, e.g., BCC from other diseases; SCC from precancerous AK; and melanoma from BMN. Unlike the previous classification task with over 150 classes, we now face a classification problem with only two alternatives. HistoGPT can be conditioned on the relevant subset of diagnoses obtained from prior knowledge or preselection. It then automatically calls a lightweight binary classifier to solve the task at hand (called "Classifier Guidance", see Methods), overcoming the class imbalance problem from above. With HistoGPT-M, we obtained weighted F1 scores of 98%, 87%, and 89% for the three tasks, respectively (Fig. 4c).

HistoGPT in "Classifier Guidance" mode also generalizes to previously unseen datasets. We demonstrated this by evaluating HistoGPT on five external, publicly available cohorts from different countries, scanner types, staining protocols, and medical procedures such as shave biopsies, punch biopsies, and excisional biopsies (Fig. 4d). While two of the cohorts (Linköping[36] and Queensland[37]) include a variety of dermatologic diseases, the other three cohorts (Münster-3H, CPTAC, TCGA) include only BCC or melanoma cases, but can still be used to assess the performance of HistoGPT. We report the classification accuracy for single-class datasets and the weighted F1 score for multi-class datasets. We retrained all models on the entire Munich dataset. In Münster-3H, HistoGPT-M with classifier guidance correctly identified BCC in 88% of cases (Fig. 4e), comparable to the established MIL approaches. The models also reliably discriminated melanoma from other types, with accuracies of 66% and 72% in TCGA and CPTAC, respectively, outperforming state-of-the-art MIL (Fig. 4e). In multi-class settings (Queensland with 3 classes and Linköping with 14 classes), we achieved weighted F1 scores of 83% and 65%, respectively (Fig. 4e). Thus, classifier guidance improves the effectiveness and generalizability of the model across different external cohorts. We also see a trend that HistoGPT and MIL-based methods perform well on datasets consisting mostly of the five most common diseases (Münster-3H, Queensland, and Linköping), highlighting the limitations of current deep learning techniques on samples (e.g., melanoma in TCGA and CPTAC) that were rarely seen during training.

Münster-1K contains 1000 random dermatopathology cases from the daily clinical routine of the University Hospital Münster. It is the only one of the five external cohorts to include (unstructured) pathology reports. In contrast to the Munich reports, these reports contain only the critical findings and the final diagnosis (e.g., "Lichen planus-like keratosis (regressive solar lentigo/flat seborrheic keratosis), no evidence of basal cell carcinoma in the present biopsy") and thus lack the detailed microscopic description of the Munich training set. Nevertheless, we were able to calculate how much diagnostic information HistoGPT encoded by comparing the extracted medical terms and measuring the cosine similarity as before (Fig. 4f). HistoGPT-L captured up to 61% of all biomedically relevant words, using our dermatology dictionary and the ScispaCy model (comparable to the Munich results of up to 63%), even though the ground truth was written in a completely different style and structure. HistoGPT-L also achieved high cosine similarity under BioBERT and GPT-3-ADA. Compared to a random report generated by BioGPT-1B, and a grounded report given by GPT-4V, the text quality of these models was consistently lower compared to all HistoGPT models (Fig. 4f). Looking at the results from all cohorts (Fig. 4b, e, f), we see that increasing the size of the language module (from BioGPT-S to BioGPT-M) has only a small effect on

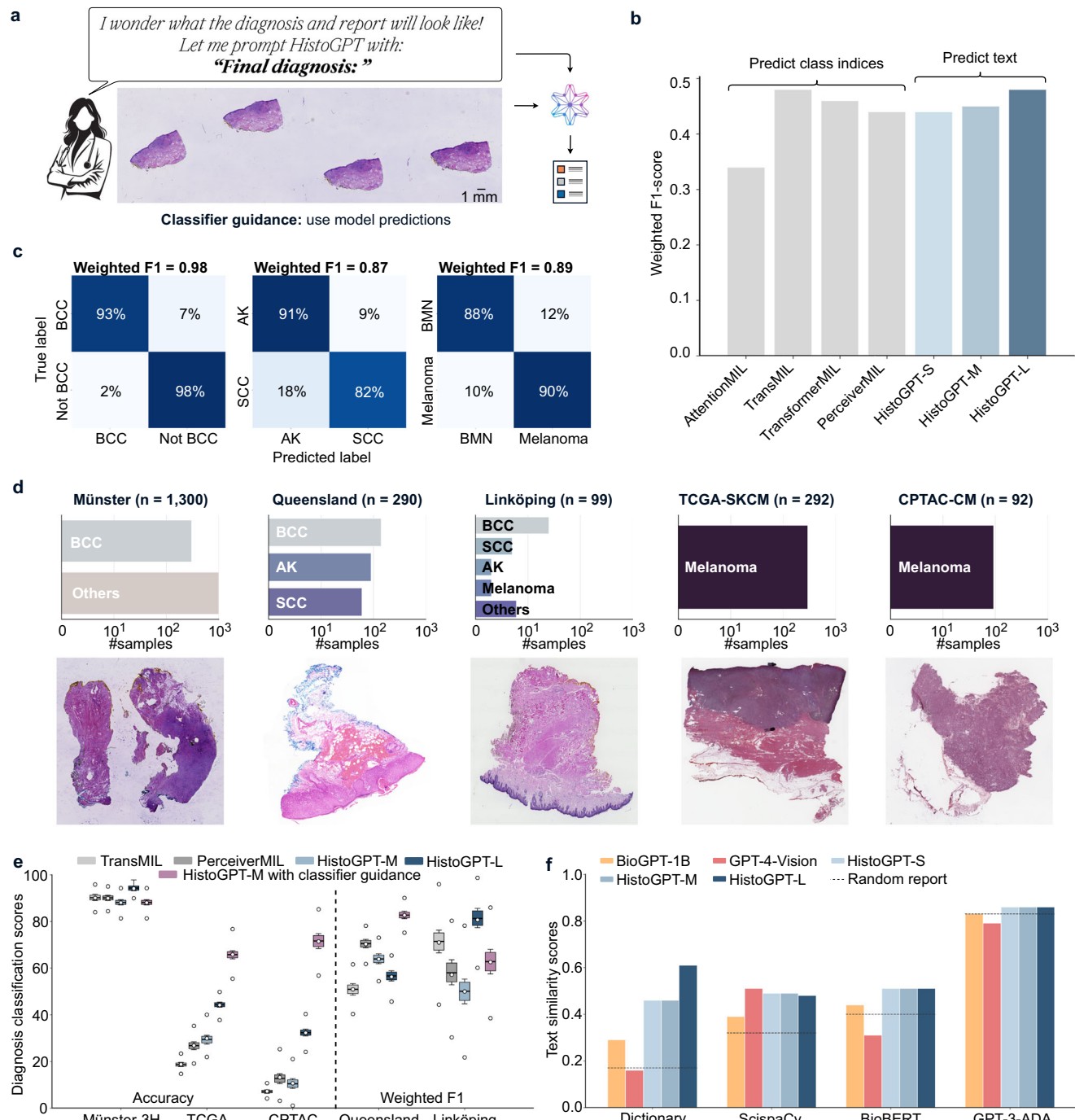

**Fig. 4 | HistoGPT accurately predicts diseases in-domain and out-of-domain without human guidance. a** In the absence of a human-in-the-loop, HistoGPT predicts the patient's diagnosis on its own and generates the corresponding pathology report. **b** On the Munich test set, HistoGPT was on par with state-of-the-art classification models in predicting over 100 dermatological diseases, even though the model's output is pure text. **c** HistoGPT discriminated malignant from benign conditions with high accuracy on the Munich dataset: basal cell carcinoma (BCC, n = 107) vs. other conditions (n = 621) with an accuracy of 0.98 and a weighted F1 score of 0.98; actinic keratosis (AK, n = 47) vs. squamous cell carcinoma (SCC, n = 33) with an accuracy of 0.88 and a weighted F1 score of 0.87; benign melanocytic nevus (BMN, n = 86) vs. melanoma (n = 21) with an accuracy of 0.89

and a weighted F1 score of 0.89. **d** We evaluated HistoGPT in 5 independent external cohorts (Münster-3H, TCGA-SKCM, CPTAC-CM, Queensland, Linköping) covering different countries, scanner types, staining techniques, and biopsy methods. **e** HistoGPT performed equal to or better than state-of-the-art MIL on external datasets, especially when using self-prompting ("Classifier Guidance"). The box plots show the quantiles as a black line and the mean as an inner circle obtained from 1000 bootstraps. The minimum and maximum values are shown as white circles at the top and bottom. **f** HistoGPT was able to produce highly accurate pathology reports, as indicated by the high keyword and cosine-based similarity scores for Münster-1K. As in Fig. 3C, the lower baseline compares two randomly selected reports. Source data are provided as a Source Data file.

downstream performance while increasing the size of the vision module and the input resolution (BioGPT-L) improves the accuracy of the reports (Supplementary Fig. 3).

## HistoGPT predicts tumor thickness, subtypes, and margins zero-shot

In the diagnosis of skin tumors, tumor thickness and subtype classification are important elements of the final report and directly influence treatment decisions. In basal cell carcinoma, tumor thickness is measured from the stratum granulosum of the epidermis to the deepest point of the tumor in millimeters, similar to the determination of the Breslow index in melanoma. It is considered an important parameter for the therapeutic approach chosen (surgical vs. non-surgical)[38]. Subtype classification based on WHO guidelines further refines treatment decisions by identifying tumor behavior and aggressiveness[39].

HistoGPT can predict both tumor thickness and tumor subtypes and does not require additional reconfiguration or specification of tumor-specific parameters at any stage of training. Given the query "The tumor thickness is", HistoGPT produces a prediction of the depth of tumor invasion without any fine-tuning. This emergent behavior is known in the literature as zero-shot learning[40]. For the 94 samples in the internal Munich test set, where tumor thickness was included in the original report, we measured a root mean square error (RMSE) of 1.8 mm and a significant Pearson correlation coefficient $\rho$ of 0.52 ($p = 9.7 \cdot 10^{-8}$, two-sided test) (Fig. 5a). In comparison, the predictions of the slide-level contrastive baselines (see Methods for a detailed description), HistoCLIP (RMSE = 4.35 mm, $\rho = 0.006$, $p = 0.96$) and HistoSigLIP (RMSE = 3.84 mm, $\rho = 0.38$, $p = 0.002$), correlated poorly with the ground truth. The state-of-the-art patch-based contrastive baseline PLIP performed even worse (RMSE = 2.78 mm, $\rho = -0.18$, $p = 0.08$). This highlights the advantage of slide-level approaches, which aggregate all patches from a whole slide image, over patch-level approaches. We observed a similar trend in the independent Münster-3H test set for PLIP and CONCH (Fig. 5e and Supplementary Fig. 4). Using gradient-attention maps, we gain insight into the reasoning behind each output. When estimating tumor thickness, HistoGPT correctly focused on the tumor region (Fig. 4c, left). However, underestimation occurred when the model struggled to find the correct spatial orientation even though it recognized the tumor mass itself (Fig. 4c, right). This was even more pronounced for HistoGPT-L trained without a position embedder, which worked on higher resolution patches that provided more detailed but less contextual information. However, the addition of a position embedder significantly restored the model's spatial awareness (Fig. 4d and Supplementary Fig. 5), increasing the Pearson correlation to 0.74 (Munich) and 0.59 (Münster).

The zero-shot capabilities of HistoGPT extended to other downstream tasks. Basal cell carcinoma (BCC) is the most common type of malignant skin cancer. Although it is the majority class in the training set, the training set does not contain BCC subtypes as final diagnoses. Therefore, BCC subtypes could not be used as labels during supervised pre-training. This information is only implicitly available as free text hidden in the microscopic descriptions or critical findings. Interestingly, HistoGPT-M was able to extract the hidden information from the in-distribution training set of the Munich cohort and apply the acquired knowledge in the out-of-distribution test set of the Münster-3H cohort to discriminate between three major BCC subtypes ("superficial", "solid/nodular", and "infiltrating") with a weighted F1 score of 0.63 (Fig. 5e). Infiltrating BCC is important to identify in routine diagnostics, as this subtype tends to have a biologically much more aggressive growth pattern and a higher relapse rate. As shown in the gradient-attention maps (Fig. 5f), HistoGPT-M correctly attended to the relevant architectural patterns within the histology slides that are the hallmarks of each BCC subtype. In comparison, HistoCLIP and HistoSigLIP achieved weighted F1 scores of 54% and 50%, respectively.

They were worse than HistoGPT-M, especially in the identification of infiltrating BCC (Fig. 5e and Supplementary Fig. 6). The two vision language foundation models for pathology image analysis, CONCH (weighted F1 = 0.31) and PLIP (weighted F1 = 0.09) did not provide good predictions for this zero-shot classification task, predicting almost all tissue slides as superficial types.

A frequently asked critical clinical question is whether a tumor is present at the surgical margin. We extracted this information from 185 reports from the Münster-1K cohort and applied the model without any fine-tuning (Fig. 5g). HistoGPT-L's zero-shot margin detection correctly detected 76% of positive margins (recall), with 73% of those flagged as positive margins actually being positive (precision), resulting in an overall F1 score of 74% for tumor margin classification. However, the model's performance in identifying healthy margins was more modest, correctly classifying 39% of negative margins (recall), with 43% of those predicted to be negative being correct (precision), resulting in an F1 score of 41%.

## HistoGPT generalizes to a real-world, multi-center clinical study

How well do the reports generated by HistoGPT work outside of Munich? To answer this question, four board-certified dermatopathologists and two board-certified pathologists from three independent clinical institutions evaluated reports generated by HistoGPT-M and HistoGPT-L on randomly selected cases from daily routine (Fig. 6a). The institutions are the Mayo Clinic (United States), University Hospital Münster (Germany), and Radboud University Medical Center (The Netherlands). The reports generated were unguided, i.e., neither "Expert Guidance" nor "Classifier Guidance" was used. One pathologist at each institution analyzed the reports, ignoring differences in report format due to language or reporting standards. A second pathologist double-checked the results. They agreed to use the following scores to grade the reports (see Methods for details): (5) beyond expectation, (4) highly accurate, (3) generally accurate with minor variations without clinical impact, (2) partially accurate with variations that could have clinical impact, (1) minimally accurate, (0) completely inaccurate. A score greater than 2 indicates a diagnosis that is considered correct or within an acceptable range of subjectivity (Fig. 6b). All cohorts include common conditions such as actinic keratosis or basal cell carcinoma, but some classes appear in only one cohort. The Munich cohort includes all of these classes for training, with varying numbers of samples and subtypes. For example, neoplastic cases have 1554 samples across 64 diseases, with an average of 27 data points per class (Fig. 6c). Overall, HistoGPT produced accurate reports for the most common neoplastic epithelial lesions (including basal cell carcinoma, melanocytic nevus, actinic keratosis, and squamous cell carcinoma), achieving an average score of 2 or higher. Performance declined for classes with limited training data (<200 cases per class) or classes that cannot be predicted from imaging alone, such as re-excision, which require additional clinical information not available to the model.

The Mayo Clinic cohort consists of 52 randomly selected cases with 84 specimens (Supplementary Fig. 7a). According to their evaluation (Fig. 6c and Supplementary Fig. 7a), HistoGPT performed particularly well in diagnosing basal cell carcinoma (achieving a score of 5 in 24 of 25 cases) and melanocytic nevi (achieving a score of 4 in all 4 cases reported as "nevus cell nevus"). There was some variation in squamous cell carcinoma and actinic keratosis cases, with scores of 3 and 4 in 15 of 21 cases. However, non-tumor/inflammatory conditions and re-excision cases without residual tumors showed low consistency and accuracy scores, with 15 of 25 cases scoring 0 and 1. Since HistoGPT was trained with only 167 melanoma cases (Fig. 3a) and did not see all possible variations of the disease, it was expected that the 2 difficult melanoma cases (Fig. 7c) would receive a score of 0. On the other hand, the good results for melanocytic nevi show

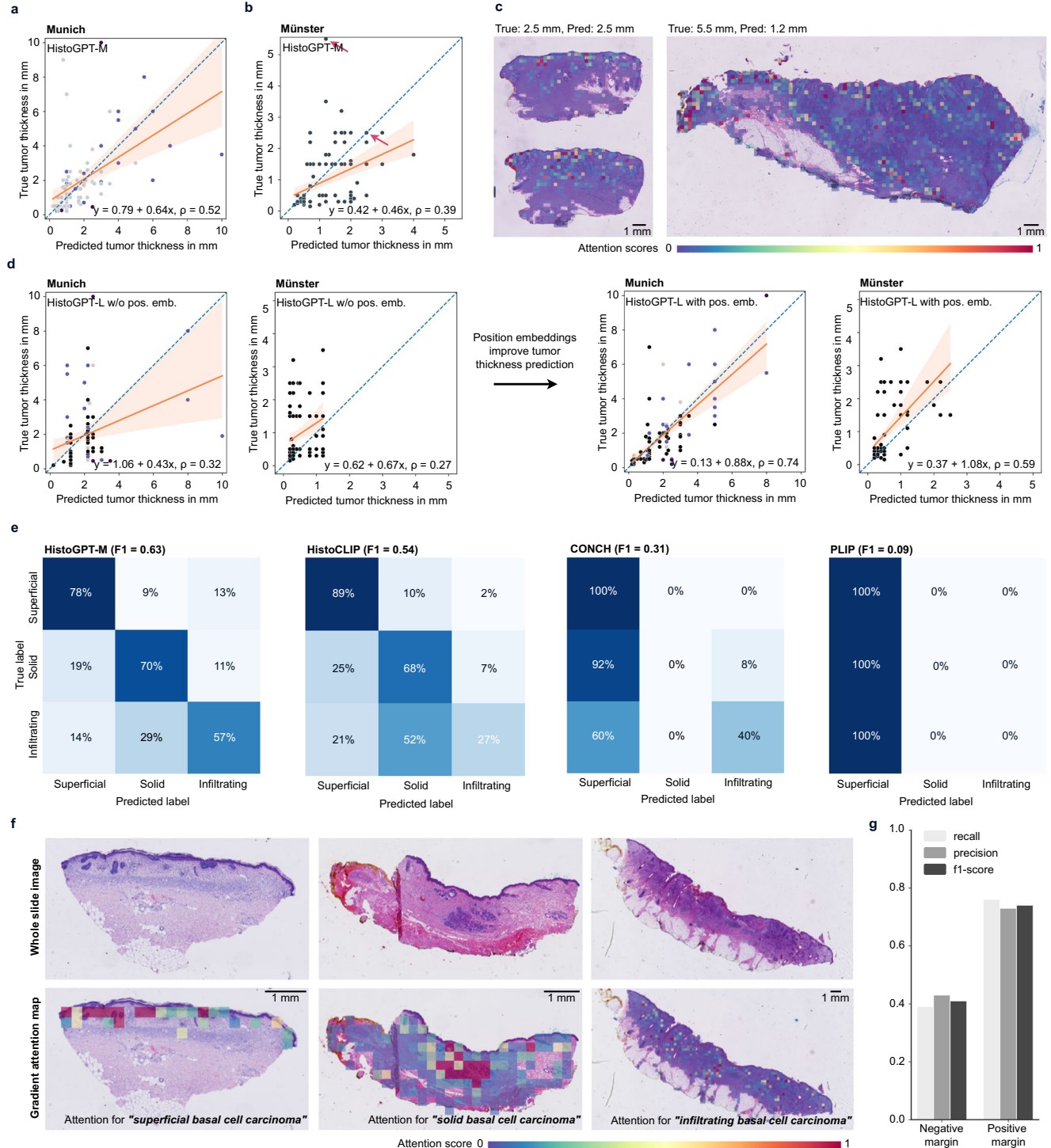

**Fig. 5 | HistoGPT predicts tumor thickness, subtypes, as well as margins in a zero-shot fashion and provides text-to-image visualization. a** HistoGPT achieved high zero-shot performance in predicting tumor thickness on the internal Munich test set. The scatter plot is color-coded according to the classes in Fig. 3a. **b** HistoGPT's prediction was also highly correlated with the ground truth on the external Münster-3H test set, even though it was obtained using a different measurement protocol. **c** Since HistoGPT is an interpretable AI system, we can understand its outputs. Here we show the two examples marked with a red arrow in this figure (**b**). Attention scores range from 0 (low attention) to 1 (high attention) as indicated by the color bar. **d** Encoding the position of each patch for the large HistoGPT model greatly improved its spatial awareness. All scatter plots include the linear regression estimate along with the 95% confidence interval as a shaded area (orange). Statistical tests were performed using a two-tailed test. **e** On the BCC subset of the independent Münster-3H cohort, HistoGPT was the only slide-level model that correctly predicted infiltrative BCC in most cases. The two patch-level models CONCH and PLIP failed in this task, predicting almost all samples as superficial. **f** Given WSIs of superficial, solid, and infiltrating BCC, HistoGPT correctly identified their morphological structures as shown by the high attention regions for the respective text strings. **g** HistoGPT predicted whether the surgical margin contained tumor or healthy cells on the out-of-distribution Münster-1K cohort without fine-tuning. Source data are provided as a Source Data file.

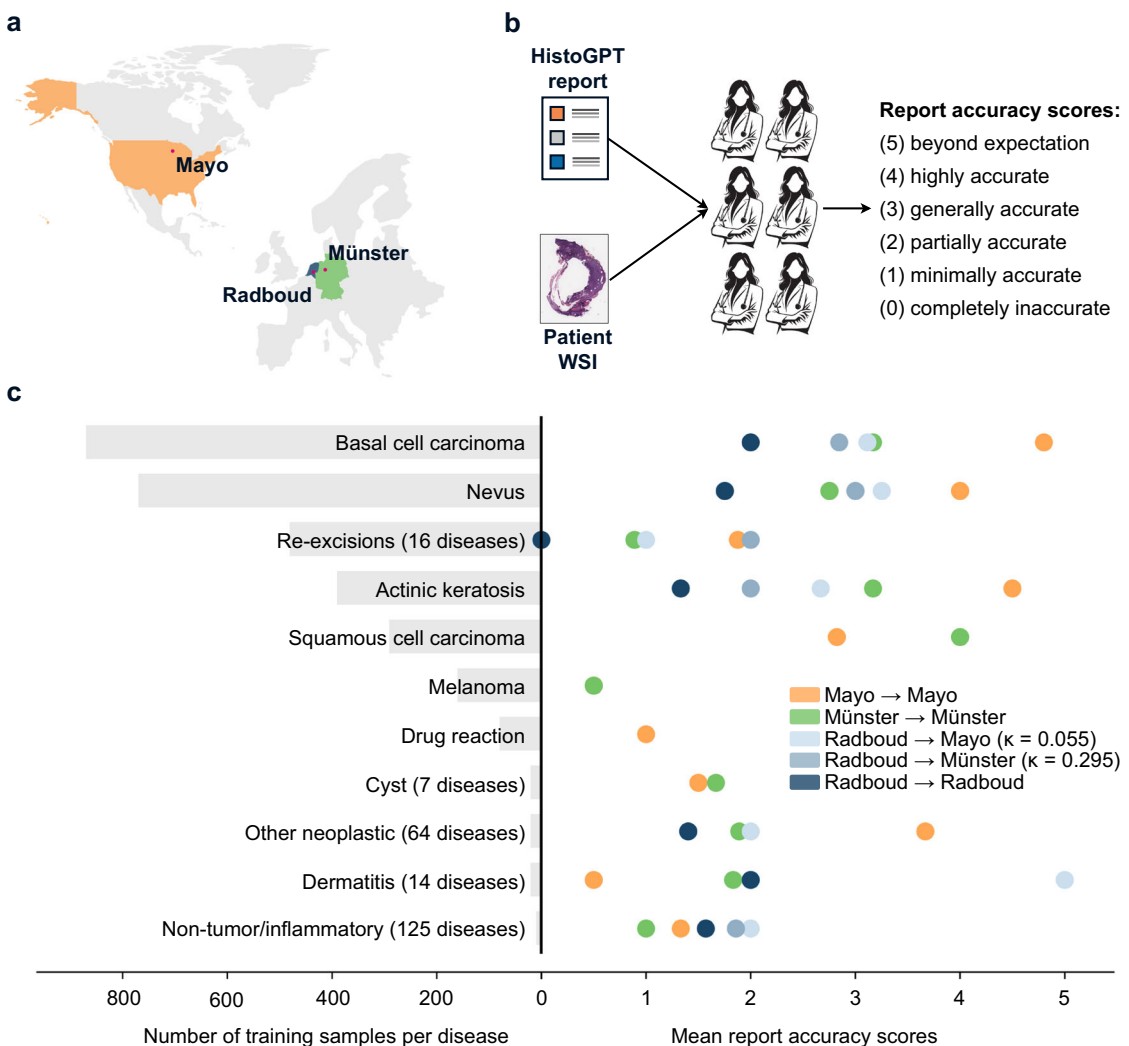

**Fig. 6 | HistoGPT produces clinically accurate and consistent pathology reports for common diseases, as confirmed in a real-world, multi-center clinical study. a** Skin biopsies were randomly collected from routine cases at the Mayo Clinic (USA), University Hospital Münster (Germany), and Radboud University Medical Center (The Netherlands). **b** Two board-certified (dermato-)pathologists at each site evaluated the generated reports according to the following criteria: (5) beyond expectation, (4) highly accurate, (3) generally accurate with minor variations without clinical impact, (2) partially accurate with variations that could have clinical impact, (1) minimally accurate, (0) completely inaccurate. A score greater than 2 indicates a diagnosis that is considered correct or within an acceptable range of subjectivity. **c** HistoGPT produced consistent and accurate reports for the most

common neoplastic epithelial lesions (achieving an average score of 2 or higher for each class) and struggled with classes with little training data (<200 data points) or classes that cannot be predicted from imaging alone (re-excisions). While our training dataset (the Munich cohort) covers all diseases provided by the three institutions, it contains many heterogeneous categories with different numbers of samples and subtypes. For instance, neoplastic cases have 1554 samples across 64 diseases, with an average of 27 data points per class. In addition, reporting standards vary widely between institutions, resulting in large variability in scores. Therefore, we also examined interobserver variability by having dermatopathologists from Mayo ($\kappa = 0.055$) and Münster ($\kappa = 0.295$) review the reports generated for the Radboud cohort. Source data are provided as a Source Data file.

that HistoGPT can detect and correctly classify melanocytic lesions as long as there are enough training samples.

The University Hospital Münster cohort is a random subset of 67 patients from Münster-1K consisting of routine cases (Supplementary Fig. 7b). Similar to the Mayo cohort, the reports generated had a score of 3 or higher (Fig. 6c and Supplementary Fig. 7b) for actinic keratosis (4 of 6 cases), basal cell carcinoma (4 of 6 cases), seborrheic keratosis (6 of 9 cases), melanocytic nevus (4 of 8 cases), and squamous cell carcinoma (2 of 2 cases). Unlike the Mayo cohort, the Münster cohort includes 9 other neoplastic conditions that were underrepresented in the training dataset (e.g., epithelioid hemangioendothelioma), which lowered the average score in this category to 2. Thus, as shown in the Mayo cohort, the performance of HistoGPT largely follows the training distribution (Fig. 3a).

The Radboud University Medical Center cohort includes 949 cases from the test split of the COBRA[41] dataset of patients who underwent skin biopsy. Two pathologists reviewed the generated reports on a subset of 50 randomly selected cases (Supplementary Fig. 7c) using the Grand Challenge platform[42]. They assigned a score of 2 or higher in 32 cases and a score of 3 or higher in 13 cases (Fig. 6c and Supplementary Fig. 7c). In contrast to the previous cohort, basal cell carcinoma only received a score of 2 in 15 of 26 cases. The lower agreement for basal cell carcinoma was due to the fact that sometimes the wrong tumor subtype was predicted (superficial instead of solid). This has clinical implications in the Netherlands, where regional guidelines recommend non-invasive treatment for superficial cases[43]. Radboud pathologists also gave lower scores because of translation errors from German to English, which resulted in incorrect or non-

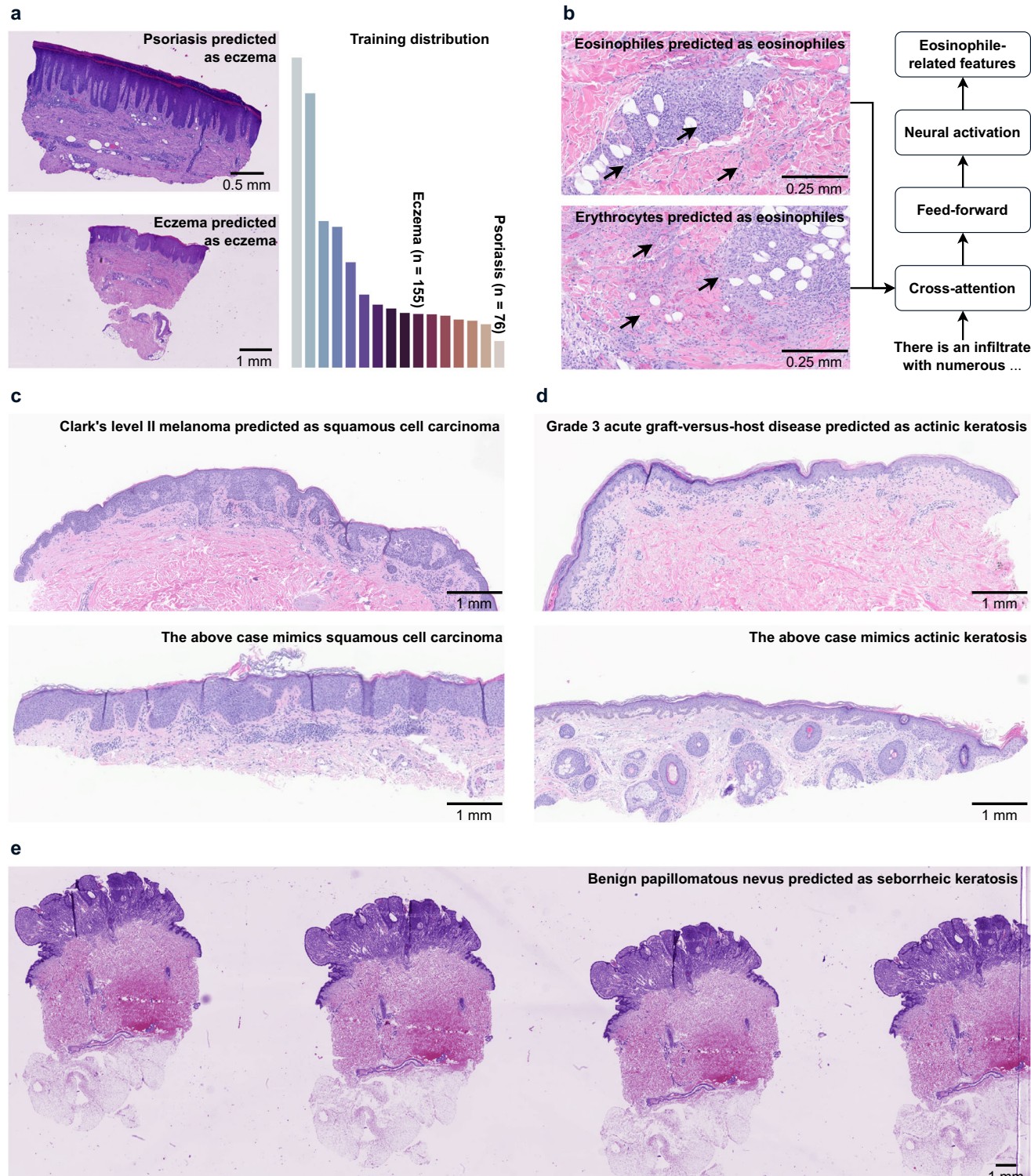

**Fig. 7 | Pathology-informed analysis of failure mechanisms. a** Like all deep learning algorithms, HistoGPT learns the most distinctive features for each class to reliably discriminate between them. However, for diseases that were rarely seen during training (e.g., psoriasis), these features are not sufficient to be applied to unseen cases and may be confused with features from related diseases (e.g., eczema). **b** Even if the cases were seen often enough during training, the tissue sample may contain tissue composition, color dynamics, and other variations that were not encountered during training. For example, we found an image of erythrocytes similar to images of eosinophils that the model saw during training, leading to the activation of eosinophil-related concepts in the neural network.

**c** Similarly, there was a case of Clark's level II melanoma (top) that mimicked the Bowenoid growth pattern of squamous cell carcinoma (bottom) and was predicted as squamous cell carcinoma. **d** Another case was a grade 3 acute graft-versus-host disease (GVHD, top) that mimicked actinic keratosis (bottom)—HistoGPT diagnosed the latter. **e** When the whole slide image contains components of different diseases, HistoGPT tends to predict the most likely diagnosis (the class seen most often during training), not the most significant one. This happened in a case of a melanocytic nevus that also showed patterns of seborrheic keratosis. Source data are provided as a Source Data file.

native medical terms (e.g., "melanocytic nevus" was often translated as "nevus cell nevus").

To measure the impact of inter-observer variability, dermatopathologists from the Mayo and Münster cohorts registered on the Grand Challenge platform and independently scored the reports generated for the Radboud cohort (Fig. 6c and Supplementary Fig. 7d). They assigned a score of 3 or higher in 27 (Münster dermatopathologists) and 33 (Mayo dermatopathologists) cases. Basal cell carcinoma also received a higher average score of 3. Fleiss' kappa values show only a slight agreement between Radboud and Mayo ($\kappa = 0.055$), and a fair agreement between Radboud and Münster ($\kappa = 0.295$) as well as Mayo and Münster ($\kappa = 0.254$). These results suggest that the lower scores assigned by Radboud pathologists may be influenced, at least in part, by subjective differences in interpretation rather than solely reflecting deficiencies in the AI-generated reports. The disagreement between pathologists highlights how inter-observer variability can influence study results and suggests that the perceived accuracy of HistoGPT reports may vary depending on the individual or institution performing the evaluation.

### Pathology-informed analysis of failure mechanisms

The three clinical evaluations confirm our previous findings that HistoGPT performs well for common diseases, but worse for rare diseases (Figs. 3a, e, 4d, e, and 6c). Thus, as with all machine learning algorithms, its quality is limited by its training data[44]—in our case, the Munich cohort. For a disease in the long tail of the training distribution, the neural network has not seen enough data points to capture all of its possible manifestations and refine its decision boundary[45]. For example, some psoriasis cases in the Münster cohort fell into the eczema category because they shared features with eczema cases that the model had previously seen during training (Fig. 7a). Notably, these two conditions can be difficult to distinguish even for dermatopathologists[46].

This, however, does not answer the question of why HistoGPT sometimes produces less accurate reports for diseases belonging to the five largest classes. The Münster dermatopathologists described the reports generated for Munich as more structured and comprehensive. Although they contained more observations than the human reports, these were not always relevant to the final diagnosis: In one case, HistoGPT mentioned a bystander cyst that was irrelevant to the diagnosis of basal cell carcinoma. In another case, the model failed to detect small objects, such as a scabies mite. On one occasion, the model incorrectly identified erythrocytes as eosinophils (Fig. 7b shows a representative example). An evaluation of the Mayo cohort showed that HistoGPT was often on the right track, but to a lesser extent than a human pathologist. In some cases, it wrote a correct microscopic description but an incorrect conclusion (either the critical findings, the final diagnosis, or both): In one case of Bowenoid squamous cell carcinoma, the microscopic description mentioned Bowenoid cells and actinic elastosis, but the critical findings concluded the presence of basal cell carcinoma. One of the most challenging cases for HistoGPT involved Clark's level II melanoma mimicking Bowenoid squamous cell carcinoma (Fig. 7c) and an acute grade 3 graft-versus-host disease (GVHD) resembling actinic keratosis (Fig. 7d). Furthermore, HistoGPT tended to hallucinate on re-excision specimens, forcing itself to make a tumor-related diagnosis when the tissue was tumor-free, which we believe can be mitigated with more contextual information. We observed a similar trend in the Münster cohort: a case of melanocytic nevus illustrates how HistoGPT correctly described the seborrheic keratosis features of a benign nevus, but completely missed the melanocytic component (Fig. 7e).

We used Ensemble Refinement to reveal the true distribution of the model for the above case (Fig. 7e). In 1 out of 10 resampled reports, HistoGPT mentioned the presence of a melanocytic nevus and made the correct diagnosis, suggesting that the model can be better

calibrated. This is consistent with our findings in the Munich and Mayo cohorts, where the model may miss the clinically relevant diagnosis on the first attempt. These limitations can be explained from first principles: HistoGPT relies on its vision module to perceive the input image, which it transforms into a series of neural activations representing high-level concepts such as diagnosis, tissue composition, or cell type[47,48]. Because HistoGPT was trained on one cohort, differences in staining protocols, scanning devices, and patient populations will inevitably shift the model's activations—a phenomenon known as batch effect. For example, the model appears to have learned that eosinophils are typically stained an intense pink and are often surrounded by tumor-infiltrating lymphocytes (Fig. 7b). When it encounters erythrocytes that appear in a similar color range and environment, the model's neural circuitry activates eosinophil-related features, leading to the prediction of eosinophils. This occurred not only in the Munich cohort during the blinded study but also in the Münster cohort during the clinical evaluation. Similarly, the cases with melanoma mimicking Bowenoid growth pattern (Fig. 7c) and GVHD resembling actinic keratosis (Fig. 7d) lead to the activation of squamous cell carcinoma and actinic keratosis-related features, respectively. We attribute this problem in part to the patch encoder, which may not be able to better discriminate some features (Supplementary Fig. 8). This is consistent with recent findings in the literature that vision language models are primarily bottlenecked by their vision module[49,50]. Inconsistencies in the reports generated are also likely due to clinical heterogeneity. For example, there was one case of seborrheic keratosis in the Münster cohort that was correctly predicted as the final diagnosis. However, the model reported verruca vulgaris, a very similar condition in this case, as the critical finding. This suggests that the image features activated highly related concepts in the cross-attention module that the language module had difficulty disentangling, leading to self-contradiction in the generated report.

## Discussion

With HistoGPT, we present a vision language model that generates pathology reports from multiple full-resolution gigapixel WSIs, e.g., from a serial section. The generated reports are highly accurate and consistent with both human reports and original specimens for the most common neoplastic diseases, as verified by an international team of six board-certified pathologists and dermatopathologists in a multi-center, multi-cohort clinical study. HistoGPT is on par with state-of-the-art multiple instance learning classification models for diagnosis prediction. It surpasses the state-of-the-art general-purpose foundation model GPT-4V, which is considered a useful tool in many clinical and pathological applications[51–53], in tissue description. HistoGPT also outperforms the state-of-the-art pathology foundation models PLIP and CONCH in zero-shot downstream tasks such as tumor thickness, subtype, and margin prediction. It is a proof-of-concept that data-driven, large-scale generative AI has great potential to assist pathologists in their clinical routine and help evaluate, report, and understand common dermatopathology cases. We have developed this model for research purposes only and as such it may not be used in patient care.

HistoGPT was trained on only 6705 clinical cases—about the number of cases a pathologist in Germany must have seen to qualify for the dermatopathology examination[54]. This number is small by LLM standards, where models are typically trained on billions of image-text pairs from the Internet. This means that HistoGPT has probably not seen enough training signals to generate detailed reports for all scenarios. For example, it performs worse on inflammatory diseases, which contain almost all minority classes, than on common diseases such as basal cell carcinoma, where even subtyping works in a zero-shot fashion. A Pearson correlation of 0.52 for tumor thickness prediction or a weighted F1 score of 0.63 for tumor subtype prediction could be improved with more training data or explicit fine-tuning for this task. Nevertheless, these results demonstrate the potential of

data-driven, large-scale generative AI, as HistoGPT was able to learn to predict tumor thickness with only 644 implicitly labeled samples.

So far, our model has only been trained and tested on dermatology cases. Therefore, it cannot yet be generalized to pan-cancer diagnosis. In addition, our training dataset suffers from severe class imbalance, which limits its usefulness for minority classes. This problem can be partially mitigated by either "Expert Guidance" or "Classifier Guidance". However, guidance also has its limitations, as the generated reports tend to be of higher quality when the model's own diagnostic prediction is also correct (Supplementary Fig. 2). While we have evaluated HistoGPT in real-world medical cohorts, only a large-scale study can confidently quantify the impact of future models on patients (like Breslow staging and clinical intervention). We believe that a future version of HistoGPT will require the use of reinforcement learning from human feedback[55] to calibrate the model's prediction to the most clinically relevant diagnosis. In addition, a mixture of train-time and test-time compute scaling along a multi-agent system will likely be required to unleash the full potential of generative AI (Supplementary Fig. 9).

## Methods

All research procedures were conducted in accordance with the Declaration of Helsinki. Ethical approval was granted by the Ethics Committee of the Technical University of Munich (reference number 2024-98-S-CB) and the Ethics Committee of Westfalen-Lippe (reference number 2024-157-b-S). The tissue samples used were from an existing biobank and were not collected specifically for this study.

As this was a retrospective data collection and the data were fully anonymized, informed consent was not obtained in consultation with the local ethics committee. No compensation was provided as there was no direct participant involvement in the study.

Sex and/or gender were not considered in the study design because the goal was to train a vision language model to generate pathology reports based solely on tissue descriptions. The input reports used to train the model did not include sex or gender information, and this information was not relevant to the study objectives. Therefore, no sex- or gender-based analysis was performed.

### Patient cohorts

**Munich cohort.** All 15,129 histology specimens from the Munich cohort were processed and stained (with hematoxylin and eosin) at the Department of Dermatology, Technical University of Munich. They were scanned with a 20× objective at 0.173 micrometers per pixel at the Core Facility Imaging at Helmholtz Munich. All slides were fully anonymized. One hundred random cases are provided in the Supplementary Material along with the reports. An example report (translated from German to English using a machine translation model) reads: "Final diagnosis: Scar. Microscopic findings: A wedge-shaped excidate with compact massive orthohyperkeratosis, focally regular acanthosis of the epidermis with hypergranulose, focally clearly flattened epidermis with elapsed reticles is presented. Underneath densely packed, partly hypereosinophilic cell-poor collagen fiber bundles, vertically placed capillary vessels. In the depth more homogenised hypereosinophilic proliferating collagen fiber bundles. Critical findings: Hypertrophic, keloid-like scar. Partial excision."

**Münster cohort.** All 1300 histologic samples of the Münster cohort were processed and stained (with hematoxylin and eosin) at the Department of Dermatology, University Hospital Münster. They were scanned with a 20× objective at 0.46 micrometers per pixel using a Hamamatsu NanoZoomer S360 MD at the Department of Dermatology, University Hospital Münster. The cohort includes 300 cases of three BCC subtypes (superficial, solid/nodular, infiltrating) with 100 samples each and 1000 cases from daily routine without special selection. All slides were fully anonymized. An example report (AI-

translated from German to English) reads: "Lichen planus-like keratosis (regressive solar lentigo/flat seborrheic keratosis), no evidence of basal cell carcinoma in the present biopsy."

**Mayo cohort.** A subset of 52 retrospective dermatopathology cases consisting of 84 specimens was randomly selected from a one-week period at the Mayo Clinic Dermatology, a tertiary medical center dermatology clinic. Cases were previously diagnosed by board-certified dermatopathologists with more than ten years of independent practice in academic centers with a digital pathology environment. Slides were scanned using a standard whole slide image scanner, and WSIs were viewed on a digital pathology image viewing platform in widespread use at the contributing authors' institution. The 52 selected cases included 50 neoplastic epithelial lesions (including basal cell carcinoma, squamous cell carcinoma, actinic keratosis, verrucous keratosis, seborrheic keratosis, inverted follicular keratosis), four cases of nevus, four cases of dermatitis, two cysts, eight re-excisions (cases with "no residual" findings), two melanomas, one case of drug reaction (with generalized pustulosis), and 13 miscellaneous/other cases, for a total of 84 specimens. An example report reads: "Skin, right melolabial fold, punch biopsy: Infiltrating basal cell carcinoma with variably clear cell features, lateral biopsy edge involved, see comment COMMENT: The carcinoma is confirmed by positivity to CK903."

**Radboud cohort.** The COBRA[41] dataset contains 5147 slides from 4066 patients. All related slides were collected from the archives of the Department of Pathology at Radboud University Medical Center, scanned with a 3DHistech Pannoramic 1000 scanner (3DHistech, Hungary) at 20× magnification (pixel resolution 0.24 μm), and subsequently anonymized. The test set was used for evaluation. It contains a total of 949 cases, with 493 non-BCC and 456 BCC samples. Non-BCC cases include patients with epidermal dysplasia (actinic keratosis or Bowen's disease) or benign conditions. Superficial BCC was observed in 24% of the slides, nodular BCC in 69%, micronodular BCC in 24%, and infiltrative BCC in 33%. For the reader study, a subset of 50 cases was randomly selected from daily routine, of which 50% were BCC (1/2 low-risk subtype, 1/2 high-risk subtype) and 50% were non-BCC cases.

### Clinical reader study

The purpose of the reader study is to simulate the use of HistoGPT in a clinical setting to measure its diagnostic accuracy and evaluate its performance on real-world medical data. All participating (dermato-) pathologists agreed on the criteria defined in the main paper. Their interpretation is discussed here. How strictly they are followed and applied to individual cases depends on the subjective judgment of each evaluator.

- We focus on key diagnoses that directly affect clinical decisions, excluding subjective subtyping unless it affects patient safety. Differences in wording and formatting are also ignored, as these vary between regions and practices. As long as the diagnosis follows the same clinical guidelines, it receives an accuracy score of 4 or 5.
- If the final diagnosis is completely wrong and has no clinical impact, it receives an accuracy score of 0 or 1. However, if the essential diagnosis is completely wrong and has a clinical impact, it is always scored as 0.
- If the diagnosis is in the correct spectrum, but there is variation in subjective assessment, it often receives an accuracy score of 3. For example, hypertrophic or bowenoid actinic keratosis vs. squamous cell carcinoma in situ.
- If the model recognizes the pattern and selects a close differential diagnosis, but not the correct diagnosis, it receives an accuracy score of 2 or 3, depending on the setting. For example, verrucous keratosis vs. seborrheic keratosis (accuracy score 3), verruca

vulgaris/HPV-associated papilloma vs. seborrheic keratosis (accuracy score 2).

## Image preprocessing

We treat all WSIs belonging to a patient as one input. In other words, we have patient-level samples instead of slide-level or even patch-level data points. For CTransPath, the WSIs were downsampled 4 times, tessellated into non-overlapping patches of $256 \times 256$ pixels, and resized to $224 \times 224$ pixels using the Python library SlideIO. UNI was trained at higher resolutions and larger patch sizes. Thus, we downsampled the WSIs 2 times and used image patches of $512 \times 512$ pixels. Background images were detected and excluded using RGB thresholding and Canny edge detection. The inputs were then converted to PyTorch tensor objects and normalized with a mean of (0.485, 0.456, 0.406) and a standard deviation of (0.229, 0.224, 0.225). We used this specific image size and normalization parameter according to the configurations of these pre-trained vision models. We typically process 1000 to 10,000 patches per slide, although this number can vary widely as some slides may contain only a few hundred spots while others may contain tens of thousands.

## Model architectures

We used CTransPath[17] (~30 million parameters) as our pre-trained vision encoder for HistoGPT-S and HistoGPT-M to extract 768-dimensional feature vectors for each image patch and concatenated them along the sequence dimension to obtain a matrix of size $n \times 768$, where $n$ is the number of image patches. The inputs are then fed into the Perceiver Resampler[26], which was originally proposed for the vision language model Flamingo[27]. We changed the default number of latents from 64 to 640 because WSIs are much larger than natural images and require a larger dimensional latent space to store the additional information. We kept the output size of 1536 because it worked well in our experiments. The fixed-size outputs of dimension $640 \times 1536$ are then used as keys and values in the tanh-gated cross-attention block (XATTN). The query vectors come from the pre-trained language model BioGPT[23] (~350 million parameters for the base model in HistoGPT-S and ~1.5 billion parameters for the large model in HistoGPT-M/L). In particular, we used one XATTN block after each language layer according to the high-performance configuration of Flamingo. The output layer of HistoGPT is a linear classifier over the vocabulary. For HistoGPT-L, we used UNI[18] (~300 million parameters) as our pre-trained vision encoder, which returns feature vectors of dimension 1024 instead of 768. We used a three-dimensional factorized position embedder adapted from NaViT[28] to encode the absolute x and y coordinates of each patch, as well as a z coordinate indicating which slide it belongs to.

Many vision language models in pathology are currently based on contrastive learning, such as PLIP and CONCH, where the embedding distance is minimized between positive pairs and maximized for negative pairs during training. However, these models have been trained on small image patches of about $224 \times 224$ pixels. Thus, they do not encode slide-level or even patient-level information and cannot be used for the downstream tasks presented in this paper (e.g., tumor subtyping and thickness prediction). We have extended these contrastive models to the slide level with HistoCLIP and HistoSigLIP and demonstrated an improvement over previous approaches. They use the feature mean of the pre-trained Perceiver Resampler as the image representation and the EOS token of the pre-trained BioGPT as the text representation. A contrastive loss then aligns both feature vectors in the common embedding space. For HistoCLIP we used the same loss as for CLIP[56]. For HistoSigLIP we used the loss proposed in SigLIP[57]. To improve performance and avoid training instabilities, we froze the vision encoder during training. This technique is called locked-image text tuning[58]). We also compared HistoGPT with the patch-based foundation models PLIP and CONCH. To aggregate the patch-level

results to the slide-level, we used majority voting and the aggregation algorithm developed for MI-Zero[59], respectively. We used PLIP as one of the current SOTA vision language models for pathology. As a vision language model, PLIP can assign a textual description to each patch. In particular, we can assign the tumor subtype prediction as a text label to each patch and then aggregate them to obtain the prediction for the whole slide. This approach was pioneered in MI-Zero. Our results show that this approach is not effective for certain tasks such as tumor subtype prediction and tumor thickness estimation. These two tasks require a comprehensive understanding of the entire tissue sample, which cannot be achieved by simply aggregating patch-level predictions. Using HistoGPT and, as an intermediate step, HistoCLIP/HistoSigLIP, we showed that a vision language model must be trained end-to-end on all patches simultaneously. This allows us to learn a slide-level representation that cannot be obtained by simply applying patch-only sampling techniques on top of existing pathology models. To further emphasize this point, we also evaluated another zero-shot foundation model for pathology, CONCH.

Since BioGPT and many other popular LLMs had all been pre-trained on mostly English literature, we needed to translate the German reports into English to take advantage of their capabilities. For the translator, we chose a standard machine translation model based on the Transformer encoder-decoder architecture[24] with the checkpoint name "Helsinki-NLP/opus-mt-de-en" available on Hugging Face.

## Model training

We pre-trained the Perceiver Resampler in a fully supervised manner by predicting the final diagnosis using a linear classifier on top of the slide encoder. Since the labels are provided at the patient level, this approach is also known as multiple instance learning (MIL). The classification head was then discarded and the resampler was plugged into the vision language model. We froze all layers of HistoGPT except the cross-attention blocks. Our generative training is based on causal language modeling: Given an input, we mask the next tokens and let the model predict them. This is done in parallel over all input tokens using an upper triangular causal attention mask.

For training, we used the AdamW optimizer with betas of (0.9, 0.95), a weight decay of 0.1, and an epsilon of 1e-8. The learning rate started at zero and warmed up linearly over 10 epochs to 1e-4 before decaying tenfold according to a cosine annealing schedule. We used a gradient accumulation of 32 to simulate a larger batch size. Each training stage consisted of 100 epochs using mixed precision training and gradient clipping to a Euclidean norm of 1.0. For contrastive learning, we used standard hyperparameters[56,57]. All models were trained on an NVIDIA A100-SXM4-80GPU on a High Performance Cluster. HistoGPT-L required 7 days of training.

During training, we randomly augmented the text inputs to avoid overfitting common words and phrases. This was done beforehand using GPT-4 to sample 9 paraphrased reports with the default temperature of 1.0 and nucleus sampling of 1.0. The prompt used was: "Rewrite the following text but be as accurate and faithful as possible to the original. Do not add or remove any information! Also, do not change the phrases 'Microscopic findings:' and 'Critical findings:', but leave them as they are."

## Classifier guidance

We enabled class imbalance awareness in HistoGPT by using a lightweight and specialized classification model. The classifier predicts one-hot encoded class indices, which are converted to text strings using a lookup table and inserted into HistoGPT. Suppose the training set contains $C$ classes. Assume that at inference time we face a classification problem with $c$ classes, where $c \subset C$. We extract features from each training sample with a pre-trained Perceiver Resampler and fit a classifier (either a linear layer or a full-sized

model) that predicts these c classes. With this approach, we reduced the 167-class classification problem to a more tractable subset of classes. For BCC vs. ¬BCC, we considered all samples that are not BCC to be ¬BCC and fitted an MLP with 100 neurons. For Melanoma vs. ¬Melanoma, we followed the same procedure. For all other classification tasks, we only trained on the specific subset. For example, if we want to classify BCC vs. SCC vs. AK vs. SK, we train a classifier only on the BCC, SCC, AK, and SK training features and ignore the remaining classes. Some datasets (Münster-3H, Queensland, and Linköping) only provide annotation masks as labels. They may contain different disease labels for different regions in the same slide. In this case, we considered the prediction of at least one of the ground truth classes to be accurate.

## Interpretability maps

To make HistoGPT more interpretable, we used partial derivatives and associated the output latents of the Perceiver Resampler with the corresponding input vectors. We then weighted the image features with the text features using the cross-attention scores. This gives us a saliency map, which we call a gradient attention map. It shows which word in the generated report corresponds to which region in a WSI. For example, we can show where the model sees basal cell carcinoma, how it detects tumor-infiltrating lymphocytes, and which regions it considers when measuring tumor thickness. In this way, we provide an approach to explainable AI by aligning visual and linguistic information.

The output of the Perceiver Resampler consists of 640 latent vectors. We computed the gradients of these latents with respect to the input patches using backpropagation. Thus, the gradient $\mathbf{G}$ has the form num_patches × num_latents. It tells us which image tokens influence which latent feature most. The mean along the latent sequence thus gives us the most important image regions according to the vision resampler. How can we use this information to determine which of these regions corresponds to which word? One idea is to give higher weights to the latents corresponding to the words we are interested in. We get these weights by looking at the cross-attention scores of the last XATTN layer. The attention matrix $\mathbf{A}$ has a dimension of num_tokens × num_latents. Thus, given a target word, we can identify the corresponding target tokens and use the corresponding rows in the attention matrix as weights. Overall, the proposed *Gradient x Attention* map is given by the weighted mean ($\mathbf{G}^T \bigcirc \mathbf{A}$[target_tokens,:].mean(dim = 0)$^T$)$^T$.mean(dim = 1).

## Evaluation metrics

We introduce two other non-trivial baselines: given the ground truth, compare two random reports with two arbitrary diagnoses (lower baseline), and compare two random reports with the same diagnosis (upper baseline). The logic behind this approach is straightforward. Medical texts often follow a structured format with a similar writing style, typically including a general description of the specimen and frequent use of common technical terms. In addition, certain diseases manifest homogeneously across patients, resulting in nearly identical report descriptions within a patient group. In such cases, the few unique terms in the reports become critical in distinguishing between different diagnoses. Therefore, these two baseline comparisons provide reference points for measuring the overall performance of our models.

Automatic evaluation of the reports generated by HistoGPT is a non-trivial task. Popular evaluation methods for natural language generation such as BLEU-4[60], ROUGE-L[61], and METEOR[62] primarily compare n-grams between two documents and may not effectively capture semantic similarities. In fact, two texts may describe the same phenomena in two different ways, making a word-by-word comparison unfair. Therefore, we focus on two different quantitative performance measures: keyword overlap and sentence similarity. For the former, we

use a comprehensive glossary of human-curated dermatological vocabularies[31] to extract important medical keywords from the ground truth notes. In addition, we use ScispaCy[32], a biomedical named entity recognition (NER) tool, to capture a broader range of technical terms. We then determine how many keywords from the ground truth text can be found in the generated text. The Jaccard index is used to quantify their overlap. To find a match in the generated report, we use an advanced version of Gestalt pattern matching (Ratcliff and Obershelp, 1988) available in the Python library difflib. We use the default cutoff threshold of 0.6. This value strikes a balance between matching every word as a target and matching only exact overlaps. The latter is undesirable because it ignores different grammatical forms of a word. As a result, some unrelated words will inevitably be matched. In this case, the Jaccard index can be considered a relative measure, since the same approach is applied to each model.

The above measures still miss some semantic nuances because certain concepts or observations (e.g., diseases, tissues, cells) may be expressed in complex phrases, possibly even involving negations. To remedy this, we use BioBERT[33] fine-tuned[63] for natural language inference and semantic textual similarity assessments. This embedding model provides the feature vectors of the generated report and the ground truth, allowing us to compute their cosine similarity as a measure of semantic understanding. To go beyond the domain-specific use of language, we apply a general large-scale embedding model, GPT-3-ADA[25], to capture a broader range of linguistic information. Similarly, we use BERTScore[64] to compute the syntactic relationship between generated and ground truth reports at the subword level.

The automatic reporting metrics above are an approximation of the actual text quality. To date, many language models have used precise word-matching metrics such as BLEU, METEOR, or ROUGE to compare two sentences. This is not appropriate for medical texts, which can be more nuanced. Our semantic-based approach better captures these nuances, but this also means that the differences between the scores are more subtle, but still important, as one or two words can completely change the clinical interpretation.

HistoGPT was trained to generate reports in the style of the Munich cohort from the Technical University of Munich. When comparing the generated report with the original reports from the in-distribution test split, a high agreement can be expected due to similar reporting standards. The out-distribution Münster cohort from the University Hospital Münster, on the other hand, contains standardized reports with smart phrases and custom templates. Therefore, the evaluation scores are not comparable on an absolute scale and should be viewed as a relative metric that compares different models on the same datasets.

For Ensemble Refinement, we summarize the bootstrapped reports by prompting GPT-4-Turbo with the instruction "Summarize the following text:". Since ER is massively time-consuming and relies on expensive API calls, we only compute the scores on a random subset of the test set (10%). However, the standard deviation among the samples remains similar to the models on the full test set, indicating that the final score would not change much. ER is closely related to boosting and ensemble learning. Therefore, we recommend using a well-based model such as HistoGPT-L over HistoGPT-S or a text-only model to turn a weak learner into a strong learner.

## Reporting summary

Further information on research design is available in the Nature Portfolio Reporting Summary linked to this article.

## Data availability

Source data are provided with this paper. The 100 patient cases from the Munich cohort used in the blinded study and the 51 patient cases from the Münster cohort for the clinical evaluation are available at

https://huggingface.co/datasets/marr-peng-lab/histogpt-dataset (DOI: 10.57967/hf/4692). The remaining samples are either publicly available at the link provided or can be requested from the original investigators: COBRA CPTAC Linköping (https://datahub.aida.scilifelab.se/10.23698/aida/drsk) Queensland (https://espace.library.uq.edu.au/view/UQ:8be4bd0) TCGA Source data are provided with this paper.

## Code availability

The code for the model can be found at https://github.com/marrlab/HistoGPT and https://zenodo.org/records/15045841 (DOI: 10.5281/zenodo.15045840)[65]. The weights for the model are available at https://huggingface.co/marr-peng-lab/histogpt (DOI: 10.57967/hf/4866). The repository is released under the Apache License 2.0, an OSI-approved open-source license. The code is freely available for use, distribution, and modification under the terms of this license. There are no access restrictions. The implementation uses the Transformers library developed by Hugging Face[66], which is also released under the Apache 2.0 License. All required attributions are included in the source code.

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

## Acknowledgements

M.T., S.J.W. and V.K. are supported by the Helmholtz Association under the joint research school "Munich School for Data Science—MUDS". C.M. acknowledges funding from the European Research Council (ERC) under the European Union's Horizon 2020 research and innovation program (Grant Agreement No. 866411 and 101113551) and support from the Hightech Agenda Bayern. We thank the dermatopathologists R. Hein, R. Franz, S. Möckel, A. Steimle-Grauer, C. Andres, and S. Roenneberg (Munich) for their detailed review of the patient material and for providing the data for this project. We also thank Nina Witte (Münster) for her technical and organizational assistance.

## Author contributions

M.T. developed the method, implemented the code, and performed the computational experiments. P.S. provided domain knowledge, analyzed the reports, and designed the medical experiments. R.G. performed the pathology analysis at the Mayo Clinic and provided additional support on clinical issues. S.J.W. supported the experiments by providing the MIL and zero-shot learning results. V.K. managed the data processing pipeline through patching and feature extraction. V.L. curated the data and trained the MIL models. B.N., D.H.M and H.D.H. were responsible for the experiments at the Mayo Clinic. J.L., M.D. and D.J.G. were responsible for the experiments at the Radboudumc. A.F. processed and scanned the WSIs for the Munich cohort. A.B. collected and annotated the images and reports for the Munich cohort. R.K. provided medical advice and designed the real-world assessment metrics. T.B. provided resources for data collection and contributed to the drafting of the manuscript. A.L.A. and A.L.M. performed the pathology analysis at the Radboudumc/Erasmusmc. F.C. and G.L. supervised the project for the Radboudumc cohort. W.C. and N.I.C. supervised the project for the Mayo Clinic cohort; N.I.C. also performed the pathology analysis at the Mayo Clinic. K.E. supervised the Munich cohort and the medical experiments. S.A.B.

supervised the Münster cohort and the medical experiments; S.A.B. also analyzed the reports in the Munich/Münster study. C.M. and T.P. supervised the machine learning experiments and the entire study.

## Funding

## Competing interests

M.T. is employed by Roche Diagnostics GmbH but conducted his research independently of his work at Roche Diagnostics GmbH as a guest scientist at Helmholtz Munich (Helmholtz Zentrum München—Deutsches Forschungszentrum für Gesundheit und Umwelt GmbH). The remaining authors declare no competing interests.

## Ethics

An interdisciplinary team of computer scientists, dermatologists, and pathologists from different institutions worked closely together. They shared their expertise and maintained the integrity of the scientific record throughout the study. Local researchers were involved in the research process to ensure that the study was locally relevant. Roles and responsibilities were agreed prior to the study and capacity-building plans were discussed. All research procedures were conducted in accordance with the Declaration of Helsinki. Ethics approval was granted by the Ethics Committee of the Technical University Munich (reference number 2024-98-S-CB) and the Ethics Committee of Westfalen-Lippe (reference number 2024-157-b-S).

## Additional information

[1]Helmholtz AI, Helmholtz Munich, Neuherberg, Germany. [2]School of Computation, Information and Technology, Technical University of Munich, Munich, Germany. [3]Department of Dermatology, Medical Center, University of Freiburg, Freiburg, Germany. [4]Department of Laboratory Medicine and Pathology, Mayo Clinic, Jacksonville, FL, USA. [5]Institute of AI for Health, Helmholtz Munich, Neuherberg, Germany. [6]MLL Munich Leukemia Laboratory, Munich, Germany. [7]Department of Quantitative Health Sciences, Mayo Clinic, Rochester, MN, USA. [8]Digital Health, Artificial Intelligence and Innovations Program, Mayo Clinic, Rochester, MN, USA. [9]Computational Pathology Group, Radboud University Medical Center, Nijmegen, The Netherlands. [10]Oncode Institute, Utrecht, The Netherlands. [11]Core Facility Pathology and Tissue Analytics, Helmholtz Munich, Neuherberg, Germany. [12]Department of Dermatology and Allergy, Technical University of Munich, Munich, Germany. [13]Department of Pathology, Radboud University Medical Center, Nijmegen, The Netherlands. [14]Department of Pathology, Erasmus University Medical Center, Rotterdam, The Netherlands. [15]Department of Dermatology and Laboratory Medicine & Pathology, Mayo Clinic, Rochester, MN, USA. [16]Dermatology Department, University Hospital Münster, Münster, Germany. [17]Department of Dermatology, Medical Faculty, Heinrich-Heine University, Düsseldorf, Germany. [18]These authors contributed equally: Manuel Tran, Paul Schmidle. [19]These authors jointly supervised this work: Kilian Eyerich, Stephan A. Braun, Carsten Marr, Tingying Peng. ✉e-mail: kilian.eyerich@uniklinik-freiburg.de; stephanalexander.braun@ukmuenster.de; carsten.marr@helmholtz-munich.de; tingying.peng@helmholtz-munich.de

