## [Peer Review File · Nature Communications]

Generating dermatopathology reports from gigapixel whole slide images with HistoGPT

Corresponding Author: Dr Carsten Marr

Version 0:

Reviewer comments:

Reviewer #2

(Remarks to the Author)

The Authors have adequately addressed my concerns. The new Figures 6 & 7 are welcome additions and are additional cohorts analysed and compared. The altered text is reasonable for my concerns. Also appreciated is the improved accessibility on github. Broadly, the Authors have demonstrated a tool workflow that produces clinically reasonable predictions of dermatopathology diagnosis and generates clinical reports that if not preferred to human-generated reports are generally non-inferior. The issues with diagnostic inaccuracy are to be expected with inflammatory disease as the supplied clinical history is well-known to be absolutely critical to making such diagnoses as the histological features are often non-specific and overlapping. In this area of dermatopathology one is in effect telling the submitting physician (dermatopathologist or otherwise) if the histologic features are compatible with their preferred diagnosis or if it favors one of a set of differential diagnoses.

In the end, this is a proof of principle project. The resulting output does not need to replace a pathologist, but needs to be helpful in automating aspects of the diagnostic process. I believe this is demonstrated in the revised submission.

I have no further comments.

(Remarks on code availability)

I only looked at ease of deployment. I think the other two authors are better qualified to review the code in more detail.

Reviewer #4

(Remarks to the Author)

Thank you for the responses provided, which I find to be satisfactory.

(Remarks on code availability)

The model weights and data are provided.

Response to Reviewer 2

Reviewer: “The Authors have adequately addressed my concerns. The new Figures 6 & 7 are welcome additions are are additional cohorts analysed and compared. The altered text is reasonable for my concerns. Also appreciated in the improved accessibility on github. Broadly, the Authors have a demonstrated a tool workflow that produces clinically reasonable predictions of dermatopathology diagnosis and generates clinical reports that if not preferred to human-generated reports are generally non-inferior. The issues with diagnostic inaccuracy are to be expected with inflammatory disease as the supplied clinical history is well-known to be absolutely critical to making such diagnoses as the histological features are often non-specific and overlapping. In this area of dermatopathology one is in effect tell the submitting physician (dermatopathologist or otherwise) if the histologic features are compatible with their preferred diagnosis or if it favors one of a set of differential diagnoses.”

Our response: We sincerely thank the reviewer for their thoughtful assessment and acknowledgement of our revisions. We are pleased to hear that the inclusion of Figures 6 and 7 effectively addresses their concerns. We also appreciate their recognition of our improved code accessibility and text presentation. We have shown that HistoGPT is generally non-inferior or even superior to human-written reports for common and non-inflammatory diseases, but struggles in more challenging cases involving inflammatory conditions without additional context such as clinical history – which is consistent with the reviewer's observations. The reviewer's insightful perspective reinforces the practical utility of our workflow, most importantly, in assisting dermatopathologists by highlighting whether the histologic features are compatible with their preferred diagnosis or whether the model favors another diagnosis.

Reviewer: “In the end, this is a proof of principle project. The resulting output does not need to replace a pathologist, but need to be helpful in automating aspects of the diagnostic process. I believe this is demonstrated in the revised submission.”

Our response: It is gratifying to hear that the reviewer recognizes the intended purpose of our work. Indeed, our primary goal was to demonstrate a proof-of-concept study for an AI-assisted workflow, not to replace pathologists altogether. We appreciate the reviewer's comments that the revised submission effectively demonstrates this point, and we have ensured that this clarification is explicitly reflected in the manuscript.

Reviewer: “I have no further comments.” “I only looked at ease of deployment. I think the other two authors are better qualified to review the code in more detail.”

Our response: We appreciate the reviewer's help and constructive feedback in improving and refining our manuscript to its current state. We are also pleased that the code is easy to use.

Response to Reviewer 4

Reviewer: “Thank you for the responses provided, which I find to be satisfactorily.”

Our response: We thank the reviewer for reading our point-by-point response and the revised manuscript, in which we have included additional cohorts (Figure 6) and an in-depth analysis of failure mechanisms (Figure 7). In summary, we have shown that HistoGPT is generally non-inferior or even superior to human-written reports for common and non-inflammatory diseases, but struggles in more challenging cases involving inflammatory conditions without clinical context. Thus, HistoGPT should be viewed as an AI-assistant that can potentially be used in the diagnostic workflow, but is by no means intended to replace human pathologists.

Reviewer: “The model weight and data are provided.”

Our response: We appreciate that the reviewer looked at our code, data, and weights, which are publicly available on GitHub and HuggingFace.